# CAN INTERPRETATION PREDICT MODEL BEHAVIOR?

## ABSTRACT

Interpretability research often aims to predict how a model will respond to targeted interventions on specific mechanisms. However, it rarely predicts how a model will respond to unseen *input data*. This paper explores the promises and challenges of interpretability as a tool for predicting out-of-distribution (OOD) model behavior. Specifically, we investigate the correspondence between attention patterns and OOD generalization in hundreds of Transformer models independently trained on a synthetic classification task. These models exhibit several distinct systematic generalization rules OOD, forming a diverse population for correlational analysis. In this setting, we find that simple observational tools from interpretability can predict OOD performance. In particular, when in-distribution attention exhibits hierarchical patterns, the model is likely to generalize hierarchically on OOD data— even when ablations show that the rule's implementation does not *rely on* these hierarchical patterns. Our findings offer a proof-of-concept to motivate further interpretability work on predicting unseen model behavior.

## 1 INTRODUCTION

When do we understand a system? One standard, that of the classic scientific method (Hepburn & Andersen, 2021), requires *testable predictions of behavior under unseen conditions*. Accordingly, interpretability research is often evaluated by predicting the effect of a test-time mechanistic intervention (Geiger et al., 2024) such as activation steering (Subramani et al., 2022; Tan et al., 2025; Liu et al., 2024) or patching (Makelov et al., 2023; Kramár et al., 2024; Todd et al., 2024; Vig et al., 2020). In contrast, interpretability researchers rarely predict the effect of a test-time *data* shift. This paper focuses on the latter objective; we seek to infer the model's rules and predict its outputs on unseen data inputs.

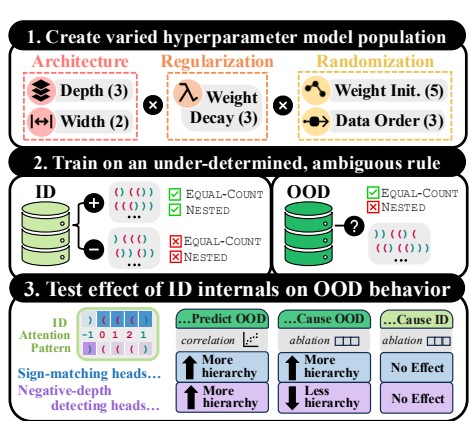

Figure 1: **Our approach.** In a population of independently trained classifiers, we correlate internal structures with OOD behaviors.

If we could predict model behavior under distribution shifts, we would unlock new interpretability applications. Well-understood instruments come with engineering tolerances corresponding to edge cases where the instrument may fail. By providing tolerances and edge cases for AI models, we might debug them, offer recommendations for reliable use, or even flag potential deployment failures. For example, structures associated with specific languages and tasks may provide clues as to whether a language model can compose them to fluently handle a given task in a particular language. Can we use interpretations to predict a model's response to unseen data?

Using a synthetic task, we show that simple analysis of attention patterns can reveal a model's generalization rule *even if* they are not used for its implementation. In other words, our interpretations are not *faithful* in a causal, mechanistic sense—but they nonetheless reveal informative traces left by the algorithm. Interpretability-based intuitions then allow us to guess what algorithm is being executed, and therefore to predict how it will treat unseen data. Current interpretability work often seeks to reverse engineer a model by identifying features that allow us to *control* it, but we look for representational structures that allow us to *simulate* it. Either aim demonstrates understanding.

We wish to predict *out-of-distribution* (OOD) model behavior by only interpreting internal model representations of *in-distribution* (ID) data. To this end, we train a large set of Transformer-based classifiers with perfect ID validation accuracy and diverse OOD behavior in a synthetic setting. Our model population is trained on data with an ambiguous classification rule: models can achieve perfect ID accuracy using either a parenthesis **counting** rule (EQUAL-COUNT) or a hierarchical parenthesis **nesting** rule (NESTED). We use an OOD test set to determine which rule each model follows. (See Appendix A for a reference glossary of terms.)

Motivated by epistemic concerns and practical, actionable interpretability objectives (Section 5), we train 270 models (Section 2.2) to classify strings of parentheses (Section 2.1). In this diverse model population, we correlate internal structure with systematic rules, finding:

- **Independently trained models cluster around systematic generalization rules.** We identify clusters of models which implement EQUAL-COUNT and NESTED by visualizing their OOD judgments (Section 3). Some unregularized training runs learn a heuristic instead of a valid rule, but only apply this heuristic OOD (Section 3.1). Upon further investigation, models can pass through transient heuristic phases in training, suggesting that vestigial circuits still affect OOD judgments. In addition to weight decay, model depth and seed influence rule learning (Section 3.2).
- **OOD generalization rules are intuitively predicted by internal representations of ID data.** We identify heads that encode hierarchical structure in their attention activations (Section 4.1). We show that models with these heads will usually apply NESTED on unseen OOD data (Section 4.2).
- **Internal structures predict generalization rules even when the rule's implementation doesn't rely on them.** We differentiate circuits that are *causally* necessary in implementing a rule from those that *correlate* with the rule across a model population. Although NESTED is associated with multiple types of hierarchical attention heads, ablation tests show that some types suppress, rather than support, the NESTED rule (Section 4.3). Furthermore, the effect of ablation is only weakly correlated between ID and OOD data conditions, calling into question the robustness of findings in causal interpretability (Section 4.3.3).

## 2 METHODS AND EXPERIMENT SETUP

To study whether we can predict OOD behavior from in distribution internals, we create models that vary in their OOD generalization. We use a training dataset compatible with at least two distinct OOD generalization rules (Section 2.1). We study the resulting variation by training a large collection of models with different hyperparameters and random seeds (Section 2.2).

### 2.1 DATA SETTING

Our setting is inspired by work on systematic generalization from ambiguous training rules (McCoy et al., 2020a;b; Murty et al., 2023). We base our dataset on the parentheses-balancing task Dyck-1 (Suzgun et al., 2019; Ebrahimi et al., 2020; Murty et al., 2023; Wen et al., 2023). Unlike standard parentheses-balancing settings, our training dataset is compatible with either EQUAL-COUNT, an unordered counting rule, or NESTED, a hierarchical parentheses-balancing rule.

Our models are classifiers, not sequence generators. They verify that the input follows some rule and output a binary class $y \in \{\text{True}, \text{False}\}$. Each input is an $n$-length sequence $s = s_1 s_2 \ldots s_n$, where $s_i \in \{(,)\}$. Using $\mathbf{1}(\cdot)$ as the indicator function, let the number of open ( and close ) tokens that appear from index 1 to $j$ be $o(j) = \sum_{i=1}^{j} \mathbf{1}(s_i = ()$ and $c(j) = \sum_{i=1}^{j} \mathbf{1}(s_i = ))$, respectively. Each rule labels $s$ as follows.

- EQUAL-COUNT is True if $s$ has the same number of open and close parentheses:

$$o(n) = c(n) \tag{1}$$

- NESTED is True if $s$ forms a recursively nested tree. All NESTED sequence are also EQUAL-COUNT, but in addition to Equation 1, a NESTED $s$ must fulfil:

$$\forall j \in \{1, \ldots, n\} \quad o(j) \geq c(j) \tag{2}$$

Our training set is compatible with both EQUAL-COUNT and NESTED: each input satisfies *both* or *neither* of Equations 1 and 2. Thus, *a model can perfectly classify ID data by either rule*. We

test which rule each model learned using an OOD set of sequences that are EQUAL-COUNT but not NESTED: they have the same number of ( and ) tokens but not in a nested order. We define accuracy according to NESTED, so a model has 100% OOD accuracy if it labels all OOD examples `False`.

**Implementation**   We create ID sequence-label pairs where NESTED and EQUAL-COUNT agree and OOD sequences where they disagree. Each sequence is generated randomly with length sampled $n \sim \text{Binomial}(40, 0.5)$ (see Appx. B). The 1K OOD examples fulfill EQUAL-COUNT but not NESTED (e.g., ))((() ). Our 1M-example train set and 1K-example ID validation set are label-balanced, containing 50% negative examples which fulfill neither EQUAL-COUNT nor NESTED (e.g., (()))) ) and 50% positive examples which fulfill both EQUAL-COUNT and NESTED (e.g., ()((()) ).

## 2.2   MODELS AND ATTENTION

We train Transformers with causal self-attention and an input length of $L = 42$. Each sequence $s$ consists of a `BOS` (beginning-of-sequence) token at $s_0$, a sequence of $n \leq 40$ parentheses, an `EOS` (end-of-sequence) token at $s_{n+1}$, and $L - n - 2$ padding tokens starting at `EOS`. We output a binary class $\hat{y} \in \{\text{True}, \text{False}\}$ at the index of the `EOS` token in the final layer.

Let $A \in \mathbb{R}^{k \times k}$ represent the attention activations of a given head on input $s$. Because our models output a classification label at the `EOS` token, we are interested in attention activations at its index $n + 1$. For index $i \in \{1, \ldots, n\}$, we therefore define $a_{\text{EOS}}(i)$ as attention to token $s_i$:

$$a_{\text{EOS}}(i) = A_{n+1,i}. \tag{3}$$

**Implementation**   We train a population of classifier models based on the minGPT architecture Karpathy (2020) with hidden dimension 64. Following Vaswani et al. (2017), we use reshaping for multi-head attention, so the model's overall parameter count is the same regardless of per-layer head count $W$. We set the learning rate $\eta = 0.0001$ with no dropout. All trained models stabilize to an ID validation accuracy of at least 99% after at most 900K training examples (Appx. Fig. 11).

We grid sweep over other hyperparameters to create a diverse model population. We train models with depths of $D \in \{1, 2, 3\}$ layers and widths of $W \in \{2, 4\}$ attention heads per layer. We also vary optimizer weight decay $\lambda \in \{0, 0.001, 0.01\}$. In each hyperparameter setting, we train models with 5 random seeds for weight initialization and 3 seeds for dataset shuffle order. This grid sweep results in 15 models per hyperparameter configuration and 270 models in total.

## 3   GENERALIZATION BEHAVIOR

All Transformer models achieve high ID validation accuracy, but their OOD behaviors vary widely.

### 3.1   MODELS CLUSTER BY SYSTEMATIC RULE

We visualize all models according to their OOD output probabilities in Fig. 2a. One cluster has near-0% OOD accuracy and another has high OOD accuracy, supporting existing claims (Qin et al., 2024; Zhao et al., 2024) that *rule-following is a categorical phenomenon, not a continuum*. Outside of these two expected EQUAL-COUNT and NESTED clusters, we find a third cluster in which models exhibit nearly identical judgments. We now characterize this cluster.

#### 3.1.1   THE HEURISTIC CLUSTER

The outlier cluster represents a simple heuristic, FIRST-SYMBOL: a sequence is `True` if its first symbol is ( and `False` if its first symbol is ). This heuristic correctly labels all positive examples, but only half of negative examples (Appx. Table 2). Although some models use this heuristic OOD, they do *not* apply it to ID data or they would not achieve high ID validation accuracy.

We hypothesize that the FIRST-SYMBOL heuristic is implemented by a vestigial circuit, which the model has learned to suppress on ID—but not OOD—data. Vestigial circuits are mechanisms required early in training which are not needed later. Prior literature (Tessier et al., 2022; Chen et al., 2023; Doshi et al., 2024) suggested that vestigial circuits are pruned by weight decay. Fig. 2b confirms that the FIRST-SYMBOL circuit only governs 1-layer models trained *without* weight decay, whereas

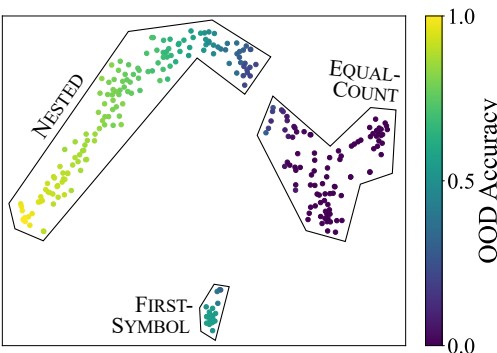
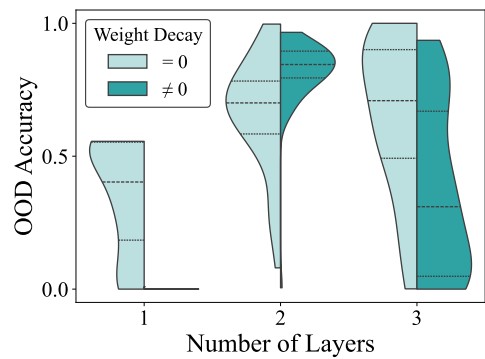

(a) T-SNE visualization of each model's output probabilities on the OOD test set. Observe three rule clusters: (1) EQUAL-COUNT with low OOD accuracy; (2) NESTED with higher OOD accuracy; (3) FIRST-SYMBOL with approx. 55% OOD accuracy.

(b) Weight decay and depth have a strong impact on OOD rule selection. The Mann-Whitney U test finds significant differences in OOD accuracy distribution between the absence and presence of weight decay across all depths ($p \ll 0.01$ across layers).

Figure 2: **Similarly trained models vary in systematic generalization rules.**

all 1-layer regularized models models follow EQUAL-COUNT. As further evidence of vestigiality, 1-layer EQUAL-COUNT models (Fig. 3) often pass through apparent heuristic phases while training, during which OOD accuracy matches that of FIRST-SYMBOL models. These findings suggest that *a vestigial circuit—which has no detectable impact on ID behavior—can still affect OOD judgments.*

### 3.2 FACTORS IN RULE SELECTION

The dominant factors in rule selection are model depth and weight decay. Appx. C details other factors, including an extension of existing findings (Abnar et al., 2020; McCoy et al., 2020a; Tran et al., 2018; Saphra & Lopez, 2020) that recurrent architectures favor hierarchical structure.

**Model depth** Overall, Fig. 2b shows that 2- and 3-layer models can learn either NESTED or EQUAL-COUNT, depending on training conditions and random variation. In contrast, 1-layer models only learn FIRST-SYMBOL or EQUAL-COUNT. Our results indicate shallower models learn simple counting rather than hierarchical rules, and that 2-layer models tend more toward hierarchy than 3-layer models. We therefore observe an inverted U-shape of hierarchical inductive bias by model depth, mirroring results from previous work Murty et al. (2023).

**Weight decay** Regularization pushes a model towards the distribution's modes and therefore

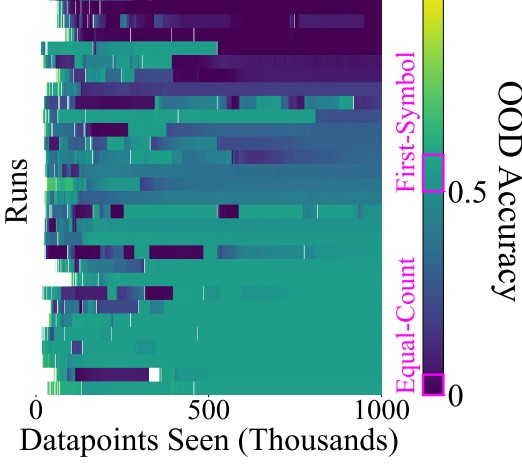

Figure 3: **Unregularized models pass through transient heuristic stages of training.** Intermediate checkpoints of unregularized one-layer models, sorted by final OOD accuracy. If a checkpoint cannot model its training distribution (ID accuracy < 0.99), we leave its cell white. The color bar is marked at FIRST-SYMBOL and EQUAL-COUNT accuracy levels. (Further analysis in Appx. D.)

to more consistently systematic rules. In particular, higher weight decay leads to more concentrated distributions of OOD behaviors across the model population (Fig. 2b). Among 1- and 3-layer models, non-zero weight decay increases the preference for EQUAL-COUNT, but among 2-layer models it increases the preference for NESTED. Although these findings support the notion that regularization promotes simple systematic rules, they complicate reductionist narratives around what rules might be described as simpler. Instead of promoting a universally simple rule, regularization promotes

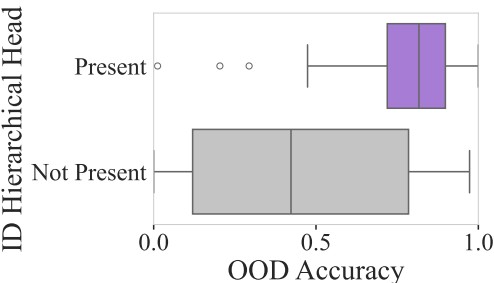

(a) Examples of attention patterns favoring violation (top) and non-violation (bottom) tokens. The input sequence is shown above the heatmaps, and each token's tree depth $c(j) - o(j)$ is displayed below.

(b) OOD accuracy of 2- and 3-layer models with and without ID hierarchical heads. Across models (and for each hyperparameter; see Appx. F), these heads correlate with OOD accuracy.

Figure 4: **Hierarchical attention patterns correlate with hierarchical generalization rules.**

different rules depending on the architectural hyperparameters. Intuitively, universal systematic rules are simpler than example memorization, but which rule is simplest—and therefore most favored by regularization—depends on the setting.

## 4   USING INTERPRETABILITY TO PREDICT BEHAVIOR

Next, we inspect the activations produced by each attention head. Using intuitive explanations of how ID inputs are processed, we predict model decisions on unseen OOD inputs. Appx. F confirms that attention patterns add predictive value even if a model's hyperparameter settings are known.

In particular, we will show that models following the hierarchical rule NESTED also display hierarchical attention patterns on ID data. These models, despite their shared outputs, can employ subtly different internal mechanisms with different causal roles. Not only do *some mechanisms lead to better OOD generalization than others*, but *some modules generalize better by maintaining the same systematic role in- and out-of-distribution*. However, hierarchical attention patterns predict that the model will follow NESTED even when these attention patterns do not directly implement NESTED.

### 4.1   HIERARCHICAL ATTENTION PATTERNS

Certain heads exhibit interpretable attention patterns at the EOS token. Specifically, these patterns indicate whether a token position $j$ follows the condition in Equation 2 that $o(j) \geq c(j)$. We will connect these intrinsically hierarchical heads to the NESTED generalization rule.

### 4.1.1   TRACKING VIOLATIONS

We identify attention patterns that encode an input's hierarchical structure. These patterns attend to a token at position $j$ depending on if it violates Equation 2, i.e., if $o(j) < c(j)$. If a sequence contains any such **violation token**, the sequence is not NESTED. In particular, *all* examples in the OOD set—composed of EQUAL-COUNT, but not NESTED), sequences—must have at least one violation token. Intuitively, then, attention patterns which track these violations can signal that a model is behaving hierarchically, closer to NESTED.

There are two ways for an attention output to **track violations** on a given sequence: a head can preferentially attend either to violation or to non-violation tokens. Given an input $s$, we say:

• An attention head favors *violation tokens* on input $s$ if there exists threshold $t > 0$ such that:

$$\forall j \in \{1, \ldots, n\}: \quad a_{\text{EOS}}(j) \geq t \text{ iff } o(j) < c(j) \tag{4}$$

• An attention head favors *non-violation tokens* on input $s$ if the conditions are reversed, i.e., there exists $t > 0$ such that:

$$\forall j \in \{1, \ldots, n\}: \quad a_{\text{EOS}}(j) \geq t \text{ iff } o(j) \geq c(j) \tag{5}$$

Examples of each pattern are shown in Fig. 4a. We say an attention head is **tracking violations on a given input** if it favors *either* violation *or* non-violation tokens on that input.

### 4.1.2 HIERARCHICAL HEAD TYPES

Some heads reliably track Equation 2 violations on each input, making them hierarchical heads across a dataset. A head is a **hierarchical head** on a given dataset if it tracks violations on at least 80% of **mixed sequences**—those containing both violation and non-violation tokens. Heads which track violations on ID examples typically behave as hierarchical heads OOD, but not always: 23% of ID hierarchical heads do not behave as OOD hierarchical heads. We divide hierarchical heads into two types:

- **Violation detector heads.** These heads consistently assign more attention to violations of Equation 2, i.e., positions preceded by more $)$ than $($ tokens. We say a head is a *Violation detector* on a given dataset if it favors violations on at least 80% of mixed sequences.
- **Twin-matching heads.** These heads behave differently depending on whether the final token follows Equation 2. If the final token violates Equation 2, the head will attend to violations; likewise, if the final token follows Equation 2 (as in all OOD sequences), the head will attend to non-violations. We say an attention head is a *twin-matching head* on a given dataset if it follows this rule on at least 80% of mixed sequences.

Among models with ID hierarchical heads, 81% have twin-matching heads, 9% have violation detecting heads, and 8% have both; only 2% of models with hierarchical heads have neither subtype. We therefore focus on these subtypes, which cover almost all ID hierarchical heads.

## 4.2 HIERARCHICAL ATTENTION ON ID DATA *predicts* HIERARCHICAL RULES ON OOD DATA

We return to our overarching question: Can these model internals intuitively suggest which function the model implements, thereby predicting its treatment of unseen OOD data? If we claim to understand a model, we should know its behavior under many unseen conditions. We now demonstrate that expectations based on our interpretations can, in fact, reliably predict OOD model behavior.

As seen in Fig. 4b, models which contain at least one ID hierarchical head are more likely to follow the NESTED rule. This result also holds separately across 2- and 3-layer models (Appx. F). These findings confirm our intuitive hypothesis: that ID hierarchical representational structure is associated with OOD hierarchical generalization behavior.

We find that 1-layer models, which notably *do not* learn the NESTED rule (Fig. 2b), possess no hierarchical heads. In fact, hierarchical heads do not occur in the first layer of any model—possibly explaining why our models can only learn violation-tracking if they have multiple layers.

On OOD sequences, which all end with $o(n) = c(n)$, a *twin-matching* head attends to non-violation tokens. We do not observe any *ID violation detector heads* switch to favoring non-violation tokens on OOD sequences. However, 25% of *ID twin-matching heads* do switch to consistently favoring violations OOD. Given that a mechanism can behave so differently OOD, it would be challenging to describe any internal mechanism in a way that applies across new domains. Nonetheless, these structures provide a *holistic* understanding of the algorithm implemented by the model. This inferred algorithm, unlike any mechanistic interpretations, applies across distribution shift.

## 4.3 HIERARCHICAL ATTENTION MAY NOT *cause* HIERARCHICAL RULES ON OOD DATA

Although hierarchical heads correlate with the NESTED rule, correlation alone doesn't establish them as causal mechanisms. To determine causality, we intervene on attention activations and examine how model performance responds. These experiments demonstrate that an attention pattern might correlate with a systematic rule without supporting it causally—in fact, we will see that the pattern may even, counter-intuitively, suppress the rule. Preventing models from displaying these attention patterns can thus enhance, rather than reduce, the correlated output behavior.

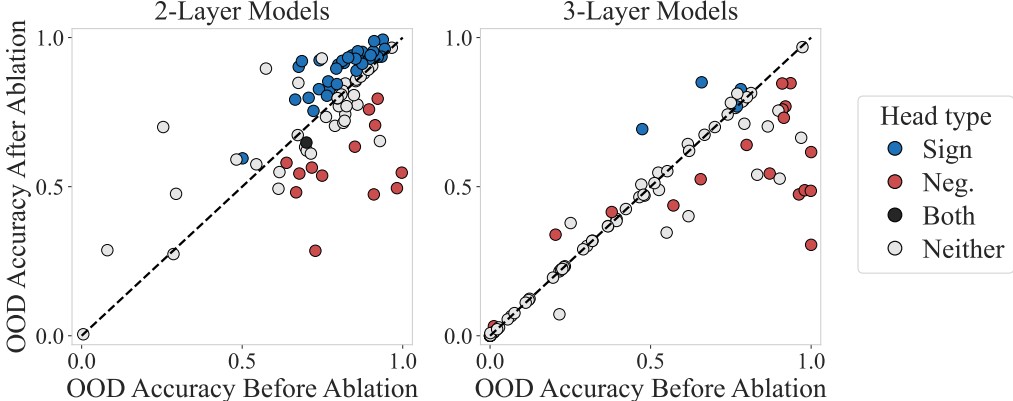

Figure 5: **Some hierarchical attention patterns damage the implementation of hierarchical rules.** OOD accuracy before and after applying uniform attention ablation to all attention heads. Each point represents a single model, colored by presence of an OOD twin-matching and/or violation detecting head. Ablation damages OOD performance in models represented by points below the diagonal, but improves OOD performance in those above the diagonal. In the former case, the attention pattern can be said to support the implementation of the NESTED rule, whereas in the latter case, the pattern suppresses the implementation of the rule. Although both head types are correlated with NESTED, only violation detection causally supports the rule.

### 4.3.1 ABLATION METHOD

To investigate whether models rely on particular attention patterns, we measure their accuracy after *uniform* attention ablation, replacing every attention activation to attend equally to all prior tokens. Our intervention preserves all other components of the Transformer and its activation, but strips any influence of violation-tracking attention or other attention activation patterns. Wen et al. (2023) demonstrated that uniform attention is sufficient to implement a parentheses-balancing task, the generative form of our NESTED classification rule. In such cases where the model does not rely on its attention patterns, replacing attention activations with uniform attention will not harm OOD accuracy, and could even improve OOD accuracy in cases where attention does not support NESTED behavior.

Note that when we apply uniform attention ablation, we apply it to *all* attention across all heads. By flattening all attention patterns, we control for all head types simultaneously and uniformly across models. Our ablation applies to attention as a whole, and is not a targeted ablation of the hierarchical heads alone. Then, one limitation to keep in mind is that some hierarchical head types may frequently co-occur with other head types that are more important. Therefore, it is difficult to solidify precise claims about causal dependencies on *specific* heads through universal uniform ablation. All heads must, however, be ablated because in some models, multiple heads are the same hierarchical type.

### 4.3.2 ABLATION RESULTS

We find that certain types of violation-tracking attention, although correlated with NESTED, actually lower the OOD accuracy of the model when present (Fig. 5). Ablating the attention of models with OOD *violation detecting* heads damages OOD accuracy, as might be expected if violation tracking is a key mechanism in implementing NESTED. By contrast, ablating the attention of models with OOD *twin-matching* heads actually *improves* OOD accuracy. The latter type of hierarchical head, although just as correlated with hierarchical generalization as the former type, is not a key mechanism in the rule's implementation. Instead, it interferes with systematic hierarchical generalization.

Our findings resist simple narratives about the mechanisms which implement a model's algorithm. Although all hierarchical heads are *correlated* with hierarchical generalization (Appx. Fig. 14), not all of them *implement* that rule under causal tests. Therefore, even simple models can be complex enough to resist causal analysis, but correlational interpretations still provide predictive value.

Unlike the hypothesised FIRST-SYMBOL circuit, OOD twin-matching heads are *promoted* by regularization, occurring more frequently when models are trained with weight decay (Appx. Fig. 13).

We therefore reject the position that twin-matching patterns are vestigial. Instead, we conjecture that these heads either develop as spandrels—side effects of learning NESTED—or they malfunction under distribution shift. Module generalization failure is a ripe topic for future work.

### 4.3.3 RESPONSE TO ABLATION IS DATA DEPENDENT

Interpretability researchers commonly test proposed explanatory mechanisms by intervening on those specific mechanisms. We call this approach into question by demonstrating that a model can *respond to an ablation on OOD data but not ID data*. Uniform attention ablation has substantial effects on OOD accuracy (Fig. 5), but leaves ID validation accuracy nearly unchanged, reduced by only 0.5% on average (Appx. H); moreover, effects of ablation on ID and OOD accuracy are only weakly correlated ($\rho = 0.24$, $p < 0.01$). These results suggest that a model might be able to compensate for the loss of a mechanism on most data, while still heavily relying on the mechanism when judging OOD edge cases. Although the model may be applying the same rule ID and OOD at the output, the distribution shift reveals brittle elements of its implementation. We posit that models could contain many redundant backup circuits that compensate for ablation ID, but become unreliable under distribution shift. Without redundancy, model judgments become more sensitive to ablations on the remaining circuits.

Based on these outcomes, a negative result from an ID causal intervention is weak evidence against a proposed mechanism. Our findings mirror several recently discovered cases where current interpretability techniques failed to generalize on novel domains. Not only are SAE dictionary features highly data-dependent (Paulo & Belrose, 2025), but steering research has shown that even features that are causal on one distribution may not be causal on new data (Kissane et al., 2024; Kantamneni et al., 2025; Smith et al., 2025). Although these findings have spurred new interest in the problem, Alvarez-Melis & Jaakkola (2018) raised the alarm about brittle interpretability years before.

## 5 PHILOSOPHICAL MOTIVATION

The interpretability of attention is subject to extensive debate (Bibal et al., 2022). Specific attention patterns are rarely necessary for model performance, either in theory (Wen et al., 2023) or in practice (Jain & Wallace, 2019; Serrano & Smith, 2019). So how do we justify our focus on attention and its link to real model behaviors? At its heart, the attention debate is a philosophical argument about what makes an interpretation faithful. We take a different epistemic position from these negative findings.

In particular, we differ in our scientific objective: predicting a target model's response to unseen inputs, rather than to mechanistic interventions. This distinction is deeply tied to concepts from the philosophy of science. The existing interpretability literature grounds notions of faithfulness in causal validity, concrete mechanisms, and implementation-level analysis. We diverge from all of these frameworks by focusing on algorithmic and computational understanding.

### INTERPRETABILITY AND CAUSALITY

In modern interpretability, causality is often held as the gold standard for evaluating understanding. Researchers causally test each module by ablating it and measuring damage to model performance. Unfortunately, the resulting findings are often compatible with multiple analyses which can only be differentiated by expensive, precise ablations. Moreover, proposed interpretations may not transfer to new settings (Stander et al., 2024; Zhang & Nanda, 2024; Makelov et al., 2023).

While interpretability demands causal tests, other sciences frequently use correlational analyses. In genetics, for example, correlational studies can link inherited traits like red hair and painkiller tolerance (Liem et al., 2005). While causal studies may seem epistemically sounder in principle, engineered approaches like single-gene editing can sabotage unrelated processes[1] through complex genetic interactions. Why should it be any easier to isolate mechanisms in highly nonlinear neural networks? Like other observational studies, we leverage correlations over model populations—not

---

[1]Haapaniemi et al. (2018) provide one famous example of how targeted gene edits can produce spurious results. They detail how CRISPR editing can automatically activate p53, a tumor suppressor. Because many edited cells die as a result, experimentalists can inadvertently select for cells with defective p53 responses, forming a confounder for research on cancer genetics.

only mechanistic interventions—to test our interpretations. In our setting, ablation tests show that some modules do not causally support generalization but nonetheless predict it (Section 4.3).

We believe interpretations of model internals can be valuable independent of causal analyses, as representational geometry may give clues as to which capabilities a model has and how it may perform under distribution shift. A model's algorithm can leave observable traces, which constitute meaningful signals regardless of their causal role. In other words, hidden representations can provide a *proxy* for the model's algorithm, even if they are not employed in its implementation.

### INTERPRETABILITY AND SCIENTIFIC REALISM

Of the many controversies in philosophy of science, few are more central than the rivalry between realism and instrumentalism (Okasha, 2016). Realists claim that the concepts used in scientific models, from quarks to gravity, are specific objects and forces acting on the world (Putnam, 1975). Instrumentalists, by contrast, argue that the epistemic goal of science is not to uncover fundamental truths about the world, but to make predictions about their observable outcomes (Duhem, 1954). To an instrumentalist, a quark might not be a real object, but simply a convenient variable in a predictive model. In the philosophy of mind, instrumentalists may even deny that internal beliefs, desires, or intentions are real phenomena, while still acknowledging that these concepts might help predict a person's behavior (Churchland, 1981; Dennett, 1991; 1989).

Our objective is an instrumentalist one. Rather than insisting on a true hierarchical mechanism, we treat hierarchical structure as part of our scientific model of network behavior. Even in a toy setting, complete causal analysis is a challenge. From an instrumentalist perspective, however, we do not need to identify the mechanisms within a model, as long as we can infer from the traces they leave.

### INTERPRETABILITY AND LEVELS OF ANALYSIS

Our understanding of a model can operate at many levels of granularity (Marr, 2010). While mechanistic interpretability often describes functions implemented by model modules, we focus on understanding the model as a whole. We aim to predict model outputs from inputs, a *computational* level—rather than *implementational*—level of analysis. In our view, full understanding requires multiple levels of analysis. We hope to see the interpretability field continue to develop through a diversity of objectives and approaches; holistic interpretability is only one under-explored direction.

## 6 DISCUSSION AND CONCLUSIONS

We show that interpretability can be used to predict model behavior on unseen inputs. If we identify similar cases in real world settings, the consequences for model evaluation could be enormous. In modern machine learning, data presents the main bottleneck to performance; by identifying edge cases where a model is likely to fail, we could efficiently evaluate model robustness.

Along with this new application of interpretability, our findings suggest a new way of evaluating interpretations: by their ability to predict model behavior on unseen inputs. We show that some attention patterns causally support OOD generalization while others limit it. By then conducting a *correlational* analysis across a model population, we discover that violation-tracking attention patterns are predictive of hierarchical generalization behavior—even when causal analysis suggests they suppress this same behavior.

These desiderata can be added to the current evaluation toolkit, which often focuses on in-distribution ablation response or correlation with data properties. As a caveat, similar analysis may be difficult in larger models because larger scales reduce—but do not remove—the impact of random variation and the isolated role of attention heads. Nevertheless, we show the instrumental, *algorithmic*-level goal of predicting model behavior is a not only worthy but also possible new objective of interpretability.

### REPRODUCIBILITY STATEMENT

All 270 models, data, and code are available at https://anonymous.4open.science/r/id-predict-ood-D6F0. Hyperparameter settings for each run are extensively documented in the main text and

appenddix of this paper. We recommend that future work experiment further with increasing scale and study the particular interactions between heads, to extend our descriptions of representation to be more granular.

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

# A  GLOSSARY

| Term | Definition and Usage |
|---|---|
| **Depth** | Parentheses sequence token chracteristic. At index $j$, the token depth is $o(j) - c(j)$ where $o(j)$ and $c(j)$ are the cumulative counts of ( and ) up to $j$. |
| **Hierarchical head** | Head that tracks a particular type of hierarchical attention pattern. Specifically, we evaluate if the EOS attention weights persistently favor either violation or non-violation tokens (e.g. Equation 4 or 5) on $\geq$80% of mixed sequences. |
| | This head type encompasses violation detectors and twin-matching heads, and it is typically found in 2- and 3-layer models. |
| **EOS** | The end-of-sequence token where attention patterns are inspected. |
| EQUAL-COUNT | Rule that a model can learn. True iff the number of open and close parentheses in a sequence are equal, regardless of their ordering (Equation 1). |
| | **Marker**: Predicts True for all OOD inputs (0% OOD accuracy). |
| FIRST-SYMBOL | Rule that a model can learn. Predicts True if first token is (, False otherwise. This rule occurs in 1 layer models with 0 weight decay. |
| | **Marker**: ~55% OOD accuracy. |
| **Depth-tracking attention** | A hierarchical attention pattern which reflects the tree structure of the preceding input sequence at an EOS token by indicating which indices violate the condition in Equation 2. The pattern can favor either violation or non-violation tokens. |
| **ID data** | In-distribution training and validation data where negative examples satisfy neither the EQUAL-COUNT nor NESTED rules. |
| **Mixed sequence** | Sequence with at least one violation and one non-violation token. Note all OOD examples are mixed. |
| NESTED | Rule that a model can learn. Returns True iff the input is properly nested (i.e, satisfies both EQUAL-COUNT and depth non-negativity, Equation 2). |
| | **Marker**: Predicts False on all OOD examples (100% OOD accuracy). |
| **Violation token** | Token position that violates the NESTED condition in Equation 2, i.e., a position preceded by more close than open symbols. |
| **Violation detector head** | Hierarchical head that preferentially attends to tokens that violate Equation 2 on $\geq 80$ % of mixed depth sequences. |
| | **Behavior**: Their presence is correlated with the hierarchical NESTED rule OOD, but ablating this head decreases models' hierarchical behavior. |
| **OOD data** | Out-of-distribution test set where negative examples satisfy EQUAL-COUNT but not NESTED. Higher accuracy is associated with NESTED. |
| **Twin-matching head** | Hierarchical head that attends to violation tokens if the last token violates Equation 2 and non-violation tokens if the last token it does not violate Equation 2, on $\geq 80\%$ of mixed sequences. |
| | **Behavior**: Their presence is correlated with the hierarchical NESTED rule OOD, but ablating this head increases models' hierarchical behavior. |
| **Attention ablation** | Flattens attention to a uniform weight distribution. Used to test causal importance of particular attention heads and patterns. |
| **Vestigial circuit** | A sub-circuit used early in training, but is not necessary for model performance at the end (e.g., FIRST-SYMBOL circuit in unregularized 1-layer models). At the end of training, such circuits may be associated with rules applied OOD but not ID. |

Table 1: Glossary of key terms used in this paper.

# B DATASET GENERATION DETAILS

We can think of the sequence length distribution as though we are generating NESTED trees with a 50% probability of recursing at each node, but discarding identical sequences. Each sequence of symbols, sampled uniformly at random, is then sorted according to which rule it follows.

## B.1 DATASET PARENTHESES SAMPLING

We create datasets as follows:

1. Sample a sequence length $n$ from a Binomial$(40, 0.5)$ distribution, with mean $20$ and variance $10$. These properties ensure that our samples are concentrated around a reasonable center, reducing extreme sequence lengths that could occur with other distributions like the Uniform. Also note that our maximum sequence length is $40$.

2. Generate a uniformly random parentheses sequence of length $n$ with the desired attributes.
   - To generate a uniformly random sequence that is neither EQUAL-COUNT nor NESTED, we choose each character independently from the set { (, ) }. If the resulting sequence satisfies EQUAL-COUNT, we discard it and generate a new one.
   - To generate a uniformly random sequence that is EQUAL-COUNT but not balanced, we randomly permute $n/2$ ( parentheses and $n/2$ ) parentheses. If the resulting sequence is NESTED, we discard it and generate a new one.
   - To generate a uniformly random sequence that is NESTED, we use the algorithm of Arnold and Sleep Arnold & Sleep (1980).

3. If the sequence generated does *not* already appear in the dataset, add it to the dataset.

Thus, each length-$n$ sequence $s$ with the desired attributes is equally likely to be chosen, and it is chosen at most once. Since we discard repeats, the empirical distribution of sequence lengths is skewed towards longer sequences, as short sequences are likely to be repeated.

We tokenize the ( and ) characters in addition to start, end, and padding tokens (BOS, EOS and PAD, respectively, with PAD appended to the end of the sequence) to ensure each sequence for classification has length $42$ (including start and end tokens). In other terms, we create a sequence of form:

$$s_0 s_1 \ldots s_n \ldots s_{41},$$

where $s_0$ is the beginning-of-sequence token BOS, $s_1$ through $s_n$ make up the $n$-length parentheses sequence $s$, $s_{n+1}$ is the end-of-sequence EOS token, and $s_{n+2}$ through $s_{41}$ are PAD tokens.

Our ID datapoints are randomly split into training and validation datasets. Each ID set contains the same number of True examples (following both EQUAL-COUNT and NESTED) and False examples (following neither EQUAL-COUNT nor NESTED). Our OOD test set consists of parentheses sequences which follow EQUAL-COUNT but not NESTED, i.e., sequences with the same number of open and closed parentheses characters, but in which the parentheses are not properly nested (ex: ))(( ). See Table 2 for examples.

Empirically, we find some models classify sequences by a FIRST-SYMBOL heuristic OOD. These models check whether $s_1 = ($ and label an OOD sequence as True if the first character is ( and False if it is ). (In-distribution, no models follow FIRST-SYMBOL, which would fail to achieve full accuracy ID.)

## B.2 RANDOM DATA ORDER IMPLEMENTATION

Our train set contains 200000 distinct datapoints. During training, we repeat this set five times, so all models were exposed to 1 million total parentheses sequences (including 5 repeats of each) across the course of training. The random seed used for data ordering, or "shuffle seed," determines the order of data within each block of 200000 training examples, so during training, a single model is exposed to the same examples in five different orderings. Models with the same shuffle seed hyperparameter encounter the 1 million total training datapoints in exactly the same order (data within each block is shuffled in a consistent way).

| Dataset | EQUAL-COUNT | NESTED | Possible $s_1$ | Example | FIRST-SYMBOL |
|---------|-------------|--------|----------------|---------|--------------|
| ID | True | True | ( | ()(()) | True |
| ID | False | False | ( 
 ) | (())() 
 )((() | True 
 False |
| OOD | True | False | ( 
 ) | ())() 
 ))() | True 
 False |
| — | False | True | — | Does Not Exist | — |

Table 2: The EQUAL-COUNT and NESTED rules applied to example parentheses sequences in our ID and OOD test sets. Also, classifications of the same parentheses sequences according to the OOD FIRST-SYMBOL heuristic. Notice that this rule does not achieve perfect accuracy ID.

## C  FACTORS IN RULE SELECTION

For consistency, we define OOD accuracy with respect to the NESTED rule. Thus a model achieving 100% OOD accuracy classifies each NON-NESTED, EQUAL-COUNT sequence in our OOD test set as False. Correspondingly, models with 0% OOD accuracy learn EQUAL-COUNT, classifying every OOD example as True (Table 2).

### C.1  ARCHITECTURE

By comparing Transformers to LSTMs, we confirm existing findings (Abnar et al., 2020; McCoy et al., 2020a; Tran et al., 2018; Saphra & Lopez, 2020) that LSTMs are intrinsically hierarchical while Transformers are not. The inductive bias of the LSTM architecture places every trained model at more than 60% accuracy on the OOD generalization set, indicating that none of these models learn EQUAL-COUNT and all are closer to the hierarchical NESTED rule. In contrast, Transformer models exhibit an OOD accuracy distribution with two peaks: one near 0% (indicating perfect application of the EQUAL-COUNT rule) and a smaller one at 90% (indicating a tendency towards NESTED). Overall, 24.4% of transformers learn EQUAL-COUNT perfectly, achieving zero OOD accuracy at the last step in training (Figure 6).

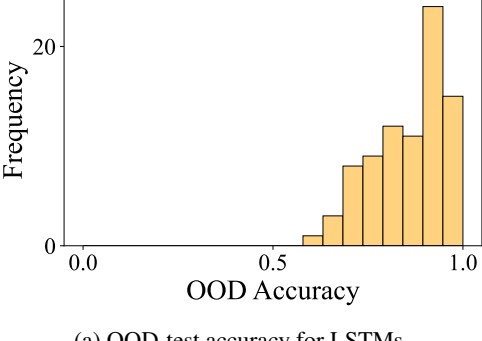
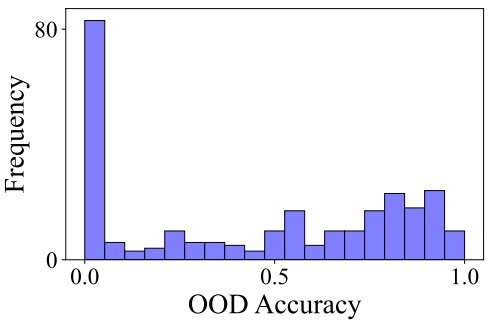

(a) OOD-test accuracy for LSTMs.   (b) OOD-test accuracy for Transformers.

Figure 6: Last OOD test accuracy for LSTM and Transformer models that achieve 99%+ ID accuracy. When LSTMs converge to near-perfect ID accuracy, they consistently also apply the NESTED rule to OOD data. Transformers, meanwhile, apply a variety of rules and exemplar-based behavior OOD.

### C.2  WIDTH

Using the non-parametric Mann-Whitney U test to detect differences between distributions, we find that the number of Transformer heads has no significant effect on the distribution of OOD accuracies for any depth of model (Figure 7). This result, which we show holds over randomness in

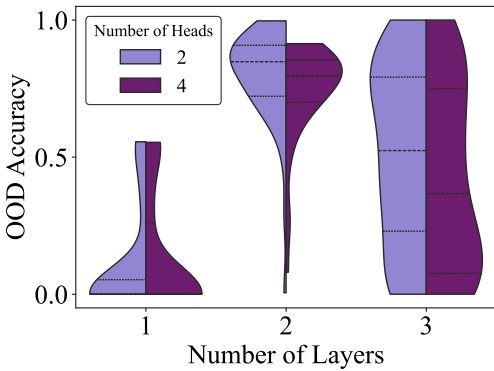

Figure 7: Unlike weight decay and depth (Figure 2b), width is not a substantial factor in final OOD rule selection in this setting. The Mann-Whitney U test finds no statistically significant differences in the distribution of last OOD accuracy over width (all $p > 0.05$).

model initialization and data exposure order, adds to a growing body of evidence across settings that changing transformer width has little effect on model expressivity and OOD generalization (Petty et al., 2024; Tay et al., 2022).

## C.3   DEPTH

In this setting, depth—unlike width—is a significant factor in rule selection, determining the peaks of the OOD behavior distribution. For instance, among the 66 out of 270 transformers (24%) which achieve perfect EQUAL-COUNT behavior with 0% OOD accuracy (Figure 6), there are 61 1-layer models, no 2-layer models, and 5 3-layer models. (Note our overall model population is evenly split into 90 each of 1-, 2-, and 3-layer models.)

We find that 1-layer models fall into a bimodal distribution centering on two rules: EQUAL-COUNT (characterized by 0% OOD accuracy) and FIRST-SYMBOL (which gives ∼55% OOD accuracy). The 12 models that generalize according to FIRST-SYMBOL all exhibit nearly identical judgments on specific examples, following a rule associated with returning a True label if the input begins with (. Because we only consider models with at least 99% validation accuracy, the models in question must also have other subroutines that are successfully applied ID but dominated by FIRST-SYMBOL OOD.

We group models via T-SNE (perplexity 12) based on their judgments on the OOD test set (Figure 2a and 9). FIRST-SYMBOL forms an outlier cluster in model judgements of 1-layer models, which otherwise primarily vary in their adherence to NESTED or EQUAL-COUNT.

By contrast, only 2% of 2-layer models are EQUAL-COUNT-leaning (determined by $< 20\%$ accuracy OOD) and none learn FIRST-SYMBOL. The mode of this model distribution is instead at 90% accuracy, firmly suggesting that most 2-layer models have approximately learned NESTED. Among 3-layer models, behavior varies enormously, with the distributional mode defined by the 20 models with $< 0.1$ OOD accuracy which learn EQUAL-COUNT. Across all 270 model training runs, 10.4% of models achieve at least 90% accuracy, including 15 2-layer and 13 3-layer models.

## C.4   REGULARIZATION

Weight decay has a significant impact on the distribution of rules models learn. Without any weight decay, models that generalize ID can converge on a variety of OOD generalization rules with wide distributions. With weight decay, particularly among smaller models, models have more similar OOD behaviors. For example, while 2-layer transformers show a consistent tendency to prefer NESTED, they achieve higher OOD accuracy more reliably when weight decay is applied (Figure 8, Figure 2b).

Among 1-layer models, training with weight decay always results in convergence to the EQUAL-COUNT rule. Without weight decay, 15.6% of 1-layer models converge to FIRST-SYMBOL with ∼55% accuracy on the OOD test set (Figure 2b). The presence of all FIRST-SYMBOL-learning models in 1-layer models with weight decay 0 indicates regularization can help prune away vestigial model

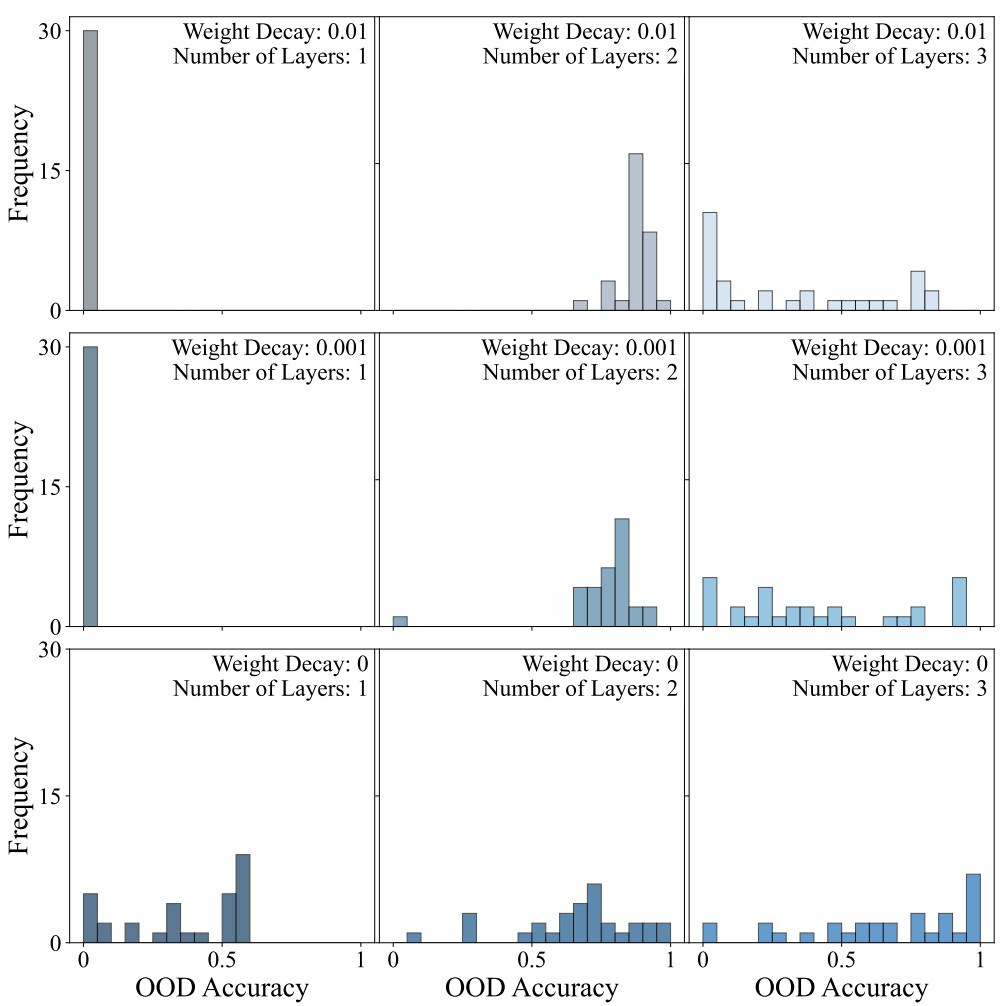

Figure 8: Final accuracy on OOD-test for Transformers of varying depths and weight decays. 1-layer Transformers learn EQUAL-COUNT or FIRST-SYMBOL. Deeper Transformers can learn EQUAL-COUNT or approximate NESTED, with 2-layer Transformers most likely to learn NESTED and 3-layer Transformers instead exhibiting more complex OOD generalization behavior.

features unnecessary for ID generalization. The presence of circuits supporting FIRST-SYMBOL may not impact ID performance, but in the absence of regularization, such features significantly decrease OOD performance.

## C.5 RANDOMIZATION IN WEIGHT INITIALIZATION AND DATA ORDER

Existing work Qin et al. (2024); Chan et al. (2022); Zhao et al. (2024); Juneja et al. (2023) has investigated the impact of random weight initialization and data order on model performance. Dodge et al. (2020) varied these factors in BERT fine-tuning, finding that modifying either factor had significant, comparable impacts on model performance. We investigate the impact of the two sources of random variation that account for differing model behaviors, and find that while they do not affect ID accuracy, both dataset ordering and model initialization affect generalization behavior.

Since we train models across three data shuffle seeds and 5 model random seeds, for a fair comparison between ranges of the impact of these factors on final OOD accuracy, we randomly select three random seeds to plot and compare the ranges of performance across shared hyper-parameter conditions.

In Figure 10, we show that in a plurality of models trained, OOD performance was impacted by at least 10%, with the maximum difference due to either one of the two random factors reaching an

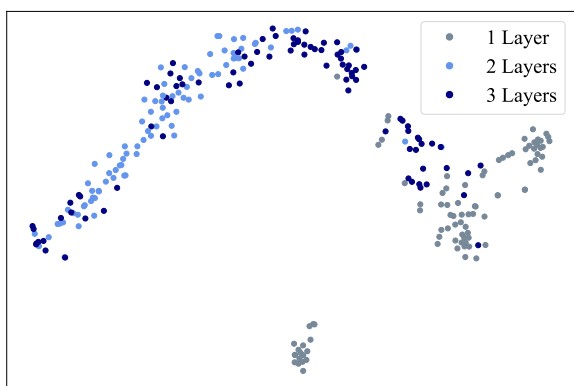

Figure 9: T-SNE of models' final OOD classifications colored by model depth.

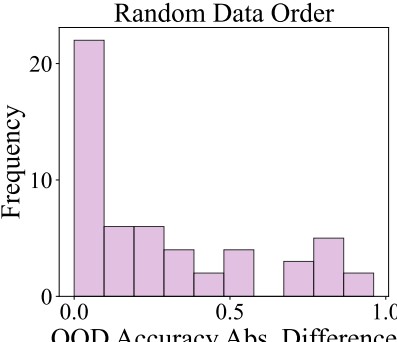
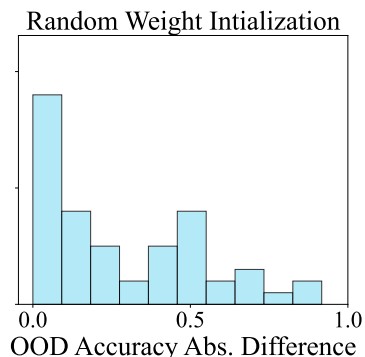

Figure 10: Data order (left) and weight initialization (right) are equally influential random factors in OOD behavior, as shown by the distribution of the ranges of final OOD accuracy between models trained with differing data ordering and weight initialization.

above 90% difference in OOD behavior. The similar distribution in the impact of random initialization and data order is aligned with previous work and indicates both factors are important to determining model OOD performance and should be accounted for in building robust ML systems.

# D  TRAINING DYNAMICS OF DIFFERENT RULES

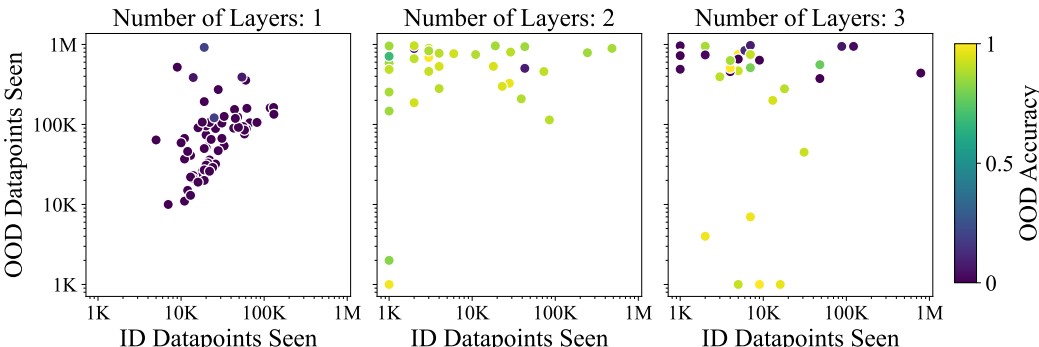

Figure 11: Illustration of generalization rules across training. ID convergence occurs when models achieve $\geq 99\%$ ID accuracy for $> 99\%$ of remaining datapoints seen—all Transformers achieve this metric after 900K datapoints. We define OOD convergence to either EQUAL-COUNT or NESTED as Transformers achieving $\leq 0.2$ or $\geq 0.8$ accuracy for $> 99\%$ of the rest of the model run, respectively, after seeing at most 975K datapoints—53% of transformers achieve this metric. Using these metrics, this plot shows the number of datapoints seen before OOD and ID convergence, excluding models that do not converge OOD.

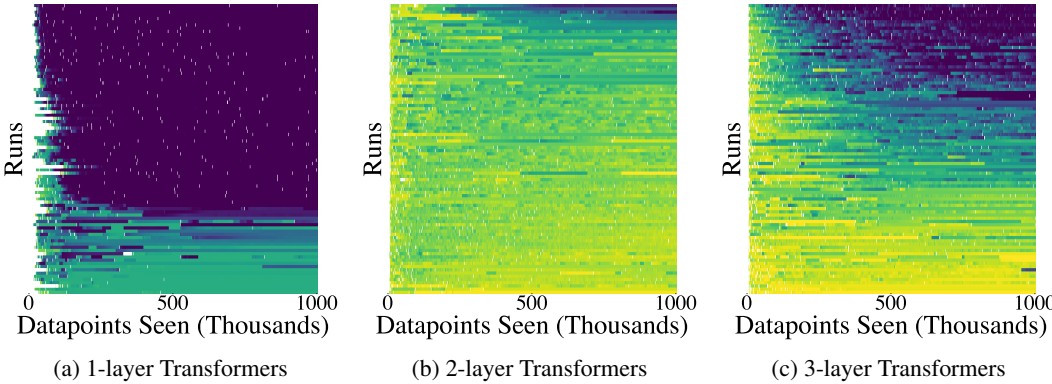

(a) 1-layer Transformers          (b) 2-layer Transformers          (c) 3-layer Transformers

Figure 12: Heatmaps showing model training dynamics broken down by depth, where purple and yellow indicates model adherence to the EQUAL-COUNT and NESTED rule, respectively (using the same scale as Figures 2a, 3, and 11). Colored cells indicate the OOD accuracy of a particular run when ID accuracy is at least 0.99.

Some OOD generalization rules can converge simultaneously with ID performance, whereas others take long to learn after the model successfully learns ID. In our setting, models can acquire and stabilize into the EQUAL-COUNT rule, but generally take longer to converge to the NESTED rule.

Although overall, models tend to classify OOD sequences as False at the outset of training—likely because the False training examples, being sampled uniformly at random, are far more diverse—they rarely stabilize immediately at high rates of False (i.e., equivalent to a NESTED rule). The models that stabilize at a NESTED rule, as seen in Figure 11, often stabilize long after ID convergence. In other words, we see an example of *structural grokking* (Murty et al., 2023). These results support the idea that NESTED is a more difficult rule to fully learn. Indeed, only three Transformer models adhere completely to the NESTED rule by classifying all OOD examples as False. The training dynamics of different rules broken down in-detail by model depth are also shown in Figure 12. Notably, deeper models have higher final OOD accuracy variance and also display greater variance during training, possibly because of their higher expressivity.

### D.1 EVALUATING THE HEURISTIC

The FIRST-SYMBOL heuristic, in contrast to the NESTED and EQUAL-COUNT rules, is not used by any models ID, where it would produce a low accuracy. However, it is used by some 1-layer models OOD, and it produces an OOD accuracy of $\approx 0.55$, reflecting the fact that around 55% of our OOD test examples begin with close brackets.

As discussed in Section 3.1.1, a majority of 1-layer models appear to pass through a FIRST-SYMBOL heuristic phase, although this heuristic does not persist until the end of training among models trained with weight decay. In this case, we say the models "appear" to pass through this phase, rather than asserting that they do, because we are only able to verify their behavior on individual datapoints at our five saved model checkpoints. However, at those model checkpoints, we are able to confirm that 1-layer models whose OOD accuracy is between $0.54$ and $0.56$ indeed almost always make OOD judgments based on first symbol. We therefore posit that throughout training, instances of OOD accuracy in this range likely reflect the FIRST-SYMBOL heuristic, though we cannot rule out that some such instances reflect some other heuristic which coincidentally produces the same OOD accuracy. Notably, 2 and 3 layer models also reach accuracies in this range while training, but, checking their at saved checkpoints, we find they do not learn FIRST-SYMBOL.

# E   ATTENTION HEADS CLASSIFIED BY OOD BEHAVIOR

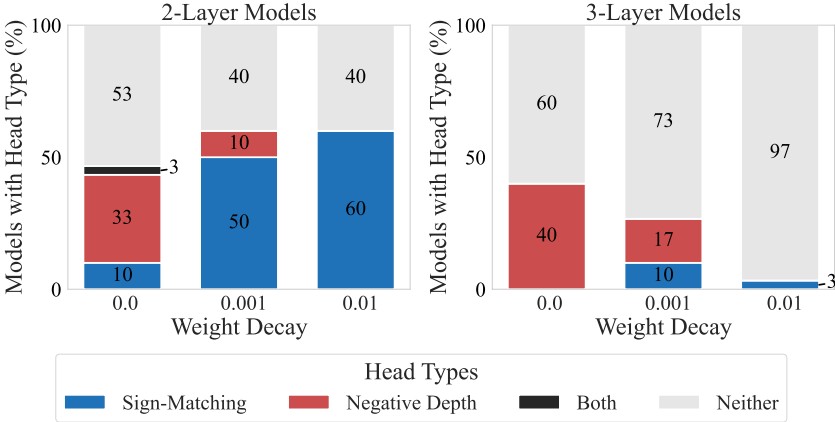

Figure 13: Percentage of 2- and 3-layer models containing each head type, by weight decay. Head types are classified according to their OOD behavior.

We breakdown the presence of types of head across 2 and 3 layer models (Figure 13).

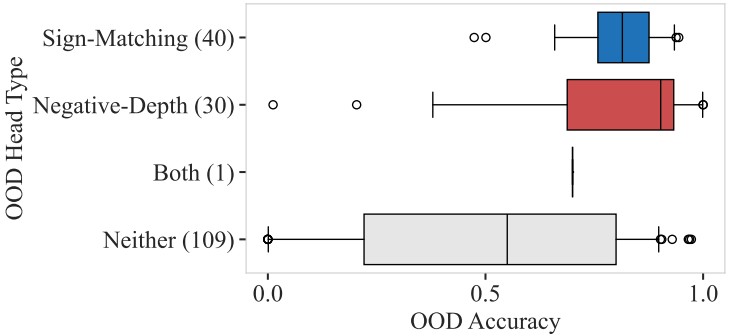

Figure 14: OOD accuracy of 2- and 3-layer models with and without OOD hierarchical heads, classified by head subtype.

In Section 4.2, we showed that classifying hierarchical heads according to their behavior on the ID validation set is predictive of generalizing according to the balanced rule. It is also possible to conduct the same analysis using behavior on the OOD test set. We find that the presence of OOD hierarchical heads is similarly predictive of generalizing according to the NESTED rule (Figure 14). All types of depth head appear to correlate similarly strongly with NESTED generalization.

# F BREAKDOWN OF HIERCHICAL HEADS BY LAYER AND WEIGHT DECAY

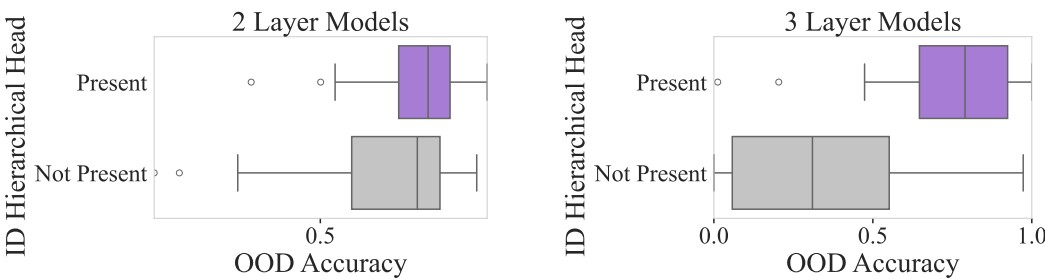

Figure 15: Last OOD test accuracy of 2 and 3-layer models with and without ID hierarchical heads.

Models with ID hierarchical heads consistently have higher OOD accuracy across both 2 and 3 layer models, indicating that model internals can provide additional predictive power to hyperparameters alone (Figure 15). Particularly for 3 layer models, which have the greatest diversity in OOD performance, the presence of ID hierarchical heads in a model provides additional insight into predicting hierarchical NESTED-like generalization.

Model internals continue to improve predictive power even when fixing both depth and weight decay simultaneously. Some hyperparameter combinations lead to one rule or the other relatively consistently: for example, "1 layer, weight decay > 0" is completely predictive of EQUAL-COUNT (see Figure 8, Appendix C.3). However, in hyperparameter settings with diverse OOD behavior, presence of hierarchical heads is predictive of NESTED generalization behavior.

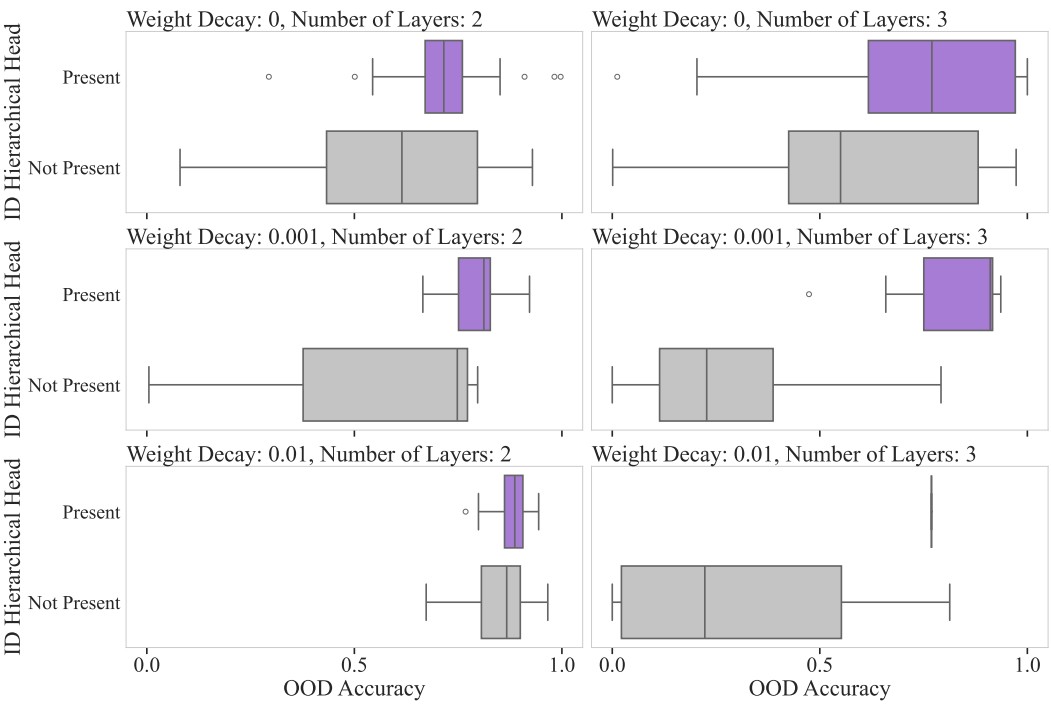

Figure 16: OOD accuracy of 2 and 3-layer models with and without ID hierarchical heads, by number of layers and weight decay value. Over the four populations, (1) 2 and 3 layer models with non-zero weight decay, (2) all 2 layer models, (3) all 3 layer models, and (4) 3 layer models with 0.001 WD, a Mann-Whitney U test shows a significant difference in OOD accuracy distributions.

In Figure 16, consider any multilayer setting producing a diverse range of OOD accuracy values, including values both above and below 50% (in other words, any setting except the "2 layer, 0.01

weight decay" setting, where all models learn NESTED). In every such setting, models without hierarchical heads have a median OOD accuracy that falls at or below the bottom quartile of the score for models with such heads. For some settings, the distributions barely overlap at all.

Thus, even if we consider the effect of hyperparameter settings, the presence of a hierarchical (i.e., depth-tracking) head is highly informative. Of course, some of these settings might lead to more hierarchical runs *because* they enable the learning of hierarchical mechanisms, meaning that the effect of hierarchical heads is far greater than any one setting would suggest.

## G    EFFECTS OF CAUSAL INTERVENTION ON ATTENTION

Our causal experiments involve uniform ablation of the attention distribution in all attention heads (Section 4.3). We ablate all attention in order to uniformly and symmetrically intervene on all models. This is in contrast to ablating exclusively hierarchical heads, which would require us to compare models on an "unequal footing," in the sense that some models would have 0 heads ablated, some models 1 head ablated, and some models 2 or more. Ablating all attention allows us to definitively eliminate all influence from hierarchical heads without introducing asymmetric interventions.

We also examined the effects of ablating individual heads one at a time. We found effects that were generally generally very similar (if weaker), in comparison with full attention ablation; in particular, in models with 2 or more hierarchical heads, full attention ablation tended to affect OOD accuracy more strongly than one-at-a-time attention ablation. Ultimately, one-head-at-a-time ablation demonstrates the same trend as full attention ablation: ablating violation heads reduces NESTED behavior, while ablating twin-matching heads increases this behavior. See Figure 17.

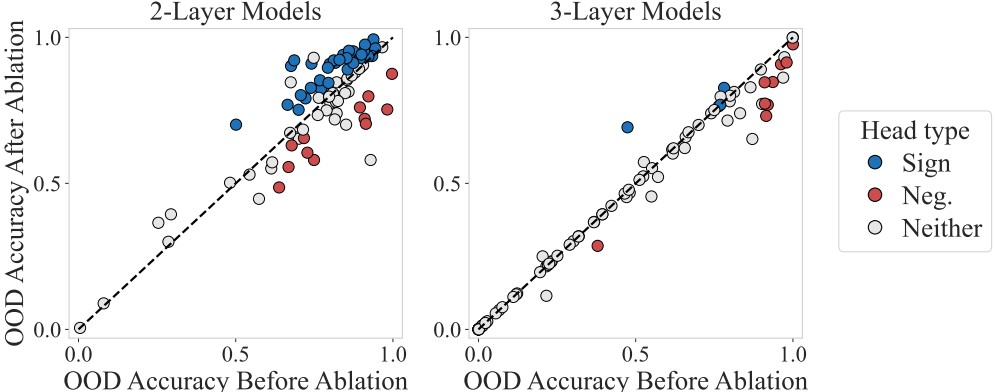

Figure 17: OOD accuracy before and after applying uniform attention ablation to one head in each model. For each model, we plot the head whose ablation affects OOD accuracy most (in absolute value), colored by whether it is a twin-matching or violation detecting head (or neither). In comparison with full attention ablation (Figure 5), effects on OOD accuracy are smaller. However, our analysis of differing effects of ablating different types of depth heads remains the same as in Section 4.3.2.

# H    CHANGE IN OOD AND ID ACCURACY AFTER ABLATING HEADS

Ablating hierarchical heads has little to no effect on ID accuracy (Figures 18 for ID hierarchical heads and 19 for OOD hierarchical heads, left panels). The maximum effect of ablating an ID or OOD head type on the ID data is 77/1000, both for violation detectors, but the median impact is around 10/1000 for all classified head types. Across ID head types, ablation tends to decrease OOD accuracy, but for OOD head types, as seen in Figure 5, ablating violation detecting heads decreases while ablating twin-matching heads increases OOD accuracy (Figures 18 and 19, right panels).

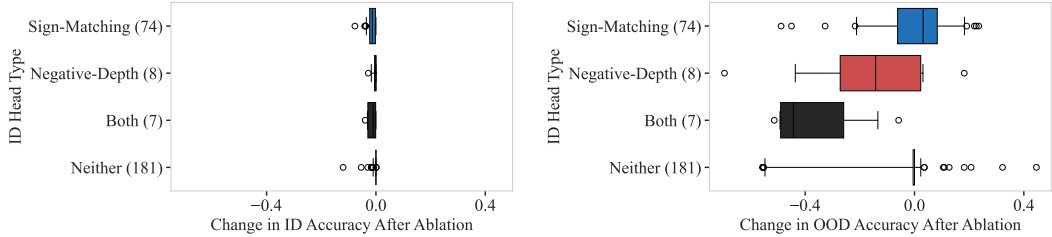

Figure 18: For heads classified by type based on their ID behavior, accuracy ID and OOD after ablation subtracted by baseline. Ablation has little impact on ID accuracy, and tends to decrease OOD accuracy across ID twin-matching and violation detecting heads. The number of models in each category are included in parentheses after the label.

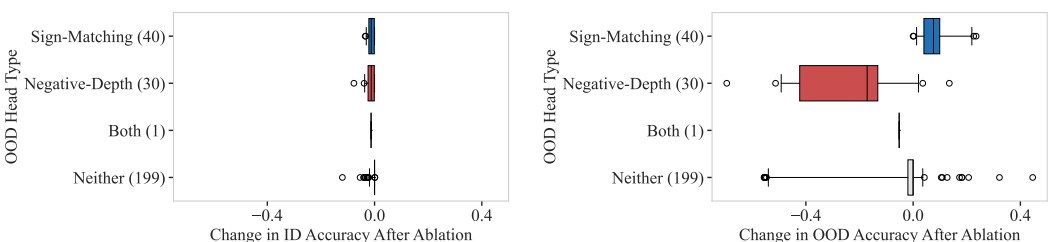

Figure 19: For heads classified by type based on their OOD behavior, accuracy ID and OOD after ablation subtracted by baseline. Ablation has little impact on ID accuracy, but ablating violation detecting OOD heads decreases and ablating twin-matching OOD heads increases OOD accuracy. The number of models in each category are included in parentheses after the label.

