# OpenReview forum: "Can Interpretation Predict Behavior on Unseen Data?"
_ICLR.cc/2026/Conference — ICLR 2026 Conference Withdrawn Submission_

### Official Review · Reviewer_ZSQM · 2025-10-26

**Soundness:** 3
**Presentation:** 3
**Contribution:** 3
**Rating:** 6
**Confidence:** 3

**Summary:**

The paper argues that by examining how Transformers allocate attention on standard (in-distribution) inputs, we can predict which rule they’ll apply to unseen (out-of-distribution) cases. Using a toy parentheses-balancing task, the authors train hundreds of small Transformers varying depth, width, regularization, and random seeds, then evaluate which rule each model actually follows OOD. They find that models cluster by their OOD rule, attention patterns can forecast that rule, and correlation does not guarantee causation.

**Strengths:**

1.	Good and ambitious motivation.
2.	Although it uses a toy setup, it includes many models and experiments, and some findings are interesting.
3.	The paper is well written, with accurate, clear descriptions and excellent details.

**Weaknesses:**

1. The experimental method relies solely on a simplified parentheses-balancing task. This narrow setup may limit the generality of the conclusions.
2. While the findings (e.g., “independently trained models cluster around systematic generalization rules”) are interesting, the paper would benefit from demonstrating at least one concrete example or use case that shows how such findings could be useful in practical applications or improvements for model designs.
3. The dataset used in evaluation is large, but I'm not entirely sure whether the empirical evidence is sufficient to support the general claims.

**Questions:**

1. Could the authors elaborate more on the design choice of using the parentheses-balancing task? Why is it an appropriate setting for testing interpretability and generalization?
2. Would the observed findings—such as the clustering of independently trained models around systematic generalization rules—also hold for other tasks or architectures beyond this synthetic setup?
3. How do the identified patterns or correlations facilitate specific downstream applications or improvements in model designs?

---

> ### Author Response · Authors · 2025-11-21
>
> > The experimental method relies solely on a simplified parentheses-balancing task. This narrow setup may limit the generality of the conclusions.
>
> We aim to highlight that an interpretation might be causally invalid but still useful for predicting model behavior. Therefore, finding any scenario where this is the case can indicate that these are distinct objectives, even if that scenario is controlled. Our primary empirical finding is that this prediction objective is decoupled from mechanistic causality, which may or may not be true in any other setting but we aim to highlight it as a real consideration when aiming to predict model behavior.
>
> > While the findings (e.g., “independently trained models cluster around systematic generalization rules”) are interesting, the paper would benefit from demonstrating at least one concrete example or use case that shows how such findings could be useful in practical applications or improvements for model designs.
>
> Our methods are not intended to be general, but to demonstrate the broader points we have about interpretability objectives. However, we are currently running experiments in a synthetic language modeling task where we will show that it is also possible to predict model behavior in autoregressive models.
>
> > The dataset used in evaluation is large, but I'm not entirely sure whether the empirical evidence is sufficient to support the general claims.
>
> Could you explain why you don’t think the empirical evidence is sufficient to support the general claims? By dataset, do you mean the training/test dataset of sequences or the number of independently trained models used for evaluating our claims?
>
> Questions:
> > Could the authors elaborate more on the design choice of using the parentheses-balancing task? Why is it an appropriate setting for testing interpretability and generalization?
>
> We use this task because it reflects existing interest in hierarchical generalization. Dyck languages represent a common task for describing model capabilities and we happened to show that they elicit a large degree of variation when models are trained as classifiers/validators rather than generators of the language.
>
> > Would the observed findings—such as the clustering of independently trained models around systematic generalization rules—also hold for other tasks or architectures beyond this synthetic setup?
>
> We show in Appendix C.1 that LSTMs can only learn NESTED and never COUNT. However, while clustering is necessary to provide a setting in which we can demonstrate our objective, it is not the primary finding of our paper. There is some prior work on clustering in other settings, such as https://arxiv.org/abs/2205.12411 and https://arxiv.org/abs/2412.04619.
>
> > How do the identified patterns or correlations facilitate specific downstream applications or improvements in model designs?
>
> If we can accomplish the proposed goal of predicting model behavior, it could enable efficient model testing by selectively testing edge cases rather than having to run a model on all deployment scenarios. The specific patterns and correlations we discuss in the paper are not general. Instead, we only want to illustrate the proposed goal and distinguish it from existing interpretability goals.

---

### Official Review · Reviewer_WV3D · 2025-10-31

**Soundness:** 2
**Presentation:** 1
**Contribution:** 2
**Rating:** 2
**Confidence:** 3

**Summary:**

This paper aims to utilize interpretability as a tool to predict model outputs on OOD, unseen input data. The problem setting explored here is a synthetic classification task on parentheses strings, where the ID training data is designed to be learnable from either an Equal-Count or Nested rule. However, the OOD test data of most interest for this problem conforms to the Equal-Count rule only; as a result, models that have internalized the Equal-Count (OOD-relevant) rule are expected to perform better on OOD inputs than those which have not. The authors demonstrate that the models indeed show evidence of learning these rules in their attention patterns (identifying several relevant design decisions), as well as other heuristics, and that there is resulting explanatory as well as predictive power from what a model has learned, specifically in predicting its performance on the (carefully defined, for this problem) OOD inputs. They also show that the common practice of ablating proposed explanatory mechanisms is data-dependent, and does not work as conventionally applied, on their OOD inputs. They claim this framework can be useful beyond the specific problem they explore.

**Strengths:**

I believe the experiments are interesting and may be sound (difficult to evaluate given the poor presentation).

Section 4, with the experiments and results around vestigial circuits and factors in rule-selection, is interesting and readable. Section 5 (also well-named) also reads better than Sections 1-3, but it was a struggle to understand the problem framing and experimental setting, so it is difficult for me to comment on soundness. The observation that the common setting of ablating proposed explanatory mechanisms is data-dependent, i.e. robust to ID data but not OOD data, would definitely be of interest to the community if the rest of the paper were more understandable. In my opinion, this last finding is the most interesting and, currently, the best-substantiated.

The figures are generally helpful in providing the appropriate intuition to understand the paper, but I would urge the authors to still be much clearer in the text.

**Weaknesses:**

MAJOR: The paper is needlessly hard to follow and feels vague in many places. It makes for a frustrated reader. Many of the questions I have about this paper are probably due to the poor exposition of the motivation, concepts, and experimental setting. The paper flip-flops awkwardly between tedious details and broad, vague conceptual or epistemic statements, making it difficult to follow, evaluate, or build on the work presented.

MAJOR: The paper is missing precise statements to guide the reader through the authors’ motivation and justification for experiments/analyses presented. For example, they should tell the reader upfront that they are first interpreting how Transformers internalize the classification rules needed to separate the training (ID) data, and then using this understanding to model behavior on OOD inputs. The current intro is disjointed and hard to follow in a first read. Without a clear outline, there’s no point in reading the subsequent sections.

MAJOR: The paper uses only a synthetic classification task on parentheses strings for validation and illustration. It is unclear how this extends to data settings where we cannot simply look at the data/task and come up with a good rule that DNNs may or may not internalize when trained. While the authors admit it is a proof-of-concept work, they do not provide any recommendation for how this might be applied in more realistic settings, where we would need to understand what heuristics/rules may apply to the underlying problem. They don’t even state what would hypothetically be necessary to generalize the methods presented. 469-470 “If we identify similar cases in real-world settings…” – how would one begin to do this?
Related to this point: 363-364 The authors state “If we claim to understand a model, we should know its behavior under many unseen conditions” but according to Table 2 on Page 15, they do not even test the very simple opposite setting of OOD data with Equal-Count false and Nested true.

The sections are also disjointed and confusingly repetitive. The two sections (3.1.1. and 3.2.1) both called Experimental Details are confusing (one is under Data and the other under Models, I guess). But the so-called “Details” sections are not actually helpful. Ideally, you have a conceptually clear description with salient details in the main text, and full details in an Appendix or later Methods section. In your case, there is actually not much detail provided, and you could collapse 3.1.1 into 3.1 and 3.2.1 into 3.2 – you’d probably save space. The paper consistently uses vague wording. Examples from the sections mentioned: “models” without specifying DNNs, then “Transformers”, then “a population of classifier models based on the miniGPT architecture with hidden dimension 64” which is still not a full description of the model (“based on”?).

(not affecting score) Why is Vaswani et al. cited with the year 2023? Everyone knows it as a 2017 paper, and the authors don’t seem to be specifically referring to any concepts from a newer version.

**Questions:**

What is the data exactly? Four paragraphs in, the authors are describing Equal-Count and Nested but have not properly explained the task setting or what the input data looks like (I see red and green parentheses in Fig 1, but no description anywhere). Even something broad like “classifying on strings” would help guide the reader–the authors can probably do better than that.

Why is the synthetic classification task not trivial? I appreciate the citations, but perhaps add a sentence about the standard settings before lines 148-149. Why Transformers needed to model this? (For example, if Transformers are being used simply for the purpose of attention-based interpretability to see whether a model has learned the logical rules of Equal-Count and/or Nested, this choice should be properly explained and justified).

In the Philosophical Motivation section, how did the authors choose which statements needed citations? No citations are provided for what they say about “other sciences” and “genetics”. In general, this section does not convincingly add value to the paper; though it attempts to situate the authors’ perspective, it is not precise or well-substantiated (in part due to the lack of citations for many of the statements given).

What does it mean for the training dataset to be “compatible” with either of the two rules? the authors repeat this several times, but the meaning doesn’t become clear because they first say:
148-150 “Unlike standard parentheses-balancing settings, our training dataset is compatible with either EQUAL-COUNT, an unordered counting rule, or NESTED, a hierarchical parentheses-balancing rule.”
and then
162-163 “Our training set is compatible with both EQUAL-COUNT and NESTED: every input sequence satisfies either both or neither of Equations 1 and 2.”
which at first seems like a contradiction: XOR vs (both or neither).
However, after re-reading the paper a few times, I think they mean to say that a model which has learned either rule (or both?) should be able to correctly classify the ID data, since all ID data points satisfy either both or neither of the constraints. In contrast, the OOD data is not classifiable from the Nested rule, only the Equal-Count rule (as in Fig 1). This is not evident from how they phrase “compatible” or "ambiguous rule” and should be properly clarified.

Where are the “OOD output probabilities” coming from? Can this be defined better since it’s so important for the clustering in Section 4?

I believe you should be clearer about stating that you train only on ID but most interested in investigating and testing on OOD.

The authors say the method is meant to be correlational and holistic. They also talk a lot about causality, and a broad conclusion of this work seems to be that even what we sometimes consider causal in XAI (interpretability ablations) is actually not strictly causal under OOD data. For a stronger paper, the authors should make more precise claims, and perhaps better separated claims, about 1) what they can interpret from an attention-based model trained in this synthetic setting where it learns hard rules or heuristics, and 2) what the implications are for current interpretability practices, from the finding that ablation analyses intended to demonstrate mechanistic causality are in fact data-dependent.

---

> ### Author Response · Authors · 2025-11-21
> **response**
>
> Thank you for this closely detailed and constructive review.
>
> > MAJOR: The paper is needlessly hard to follow and feels vague in many places.
>
> We have put effort into reducing the amount of “tedious details” required for the reader to understand the paper, so that our larger points get across better. In particular, we have changed the terminology to avoid requiring readers to think in terms of tree depth, as mentioned in our top level comment. We also address the issue of the conceptual discussion in our top level comment, and though we consider the epistemic argument to be key to our work, we have added more case study details to make the motivation less abstract.
>
> > MAJOR: The paper uses only a synthetic classification task on parentheses strings for validation and illustration.
>
> Thank you for pointing this out! While our work is focused on the objective of predicting model behavior rather than on any specific method of doing so, we believe that it could be possible to evaluate when models will fail at some point in their pipeline. For example, tasks like fact look up in different languages have been decomposed into specific pipelines which can exhibit failures at different stages. Is it possible to predict when a particular fact will fail to compose with a particular language, for example by identifying key features and modeling when they could interfere with each other? We believe there are many possible tasks where it would be productive to predict OOD behavior. Specific methods for doing so are outside of our scope.
>
> > The authors state “If we claim to understand a model, we should know its behavior under many unseen conditions” but according to Table 2 on Page 15, they do not even test the very simple opposite setting of OOD data with Equal-Count false and Nested true.
>
> As Table 2 explains, this condition is impossible because Nested is a subset of Equal-Count, as Nested strings follow the Equal-Count rule (Equation 1) and an additional rule (Equation 2). We use the abbreviation DNE, but it is possible many readers are unfamiliar with the abbreviation and so we have changed it to Does Not Exist for clarity.
>
> > The sections are also disjointed and confusingly repetitive. The two sections (3.1.1. and 3.2.1) both called Experimental Details are confusing (one is under Data and the other under Models, I guess).
>
> You correctly understood that these two different detail sections respectively discuss the data and models. We have collapsed the implementation details into their sections as suggested. We say “based on MiniGPT” because we use the MiniGPT code, but train it to produce a class label at EOS, rather than with an autoregressive next token prediction objective.
>
> > What is the data exactly?
>
> Yes, you do understand the task: We classify strings, as emphasized in section 3.1 (now 2.1). We have added this detail to the introduction in the revised introduction. The data generation process is detailed in section 3.2 (now 2.2).
>
> > Why is the synthetic classification task not trivial?
>
> We agree that the task is easily achieved by many models simpler than transformers. The reason we use this trivial task is because we want to train hundreds of small models with the same performance in distribution but different out of distribution performance. We use transformers because of how variable their behavior is in other known hierarchical generalization tasks, allowing us to study that variation. By contrast, our LSTM experiments in Appx C reflect existing understanding of their inductive bias: They only learn NESTED.
>
> > In the Philosophical Motivation section, how did the authors choose which statements needed citations?
>
> Because our primary objective is to provide an alternative goal and epistemic approach, we cannot forgo this discussion, but we have added more citations to the philosophy of science literature and cited specific case studies. As stated in our top level comment, we have moved the section to the end of the paper, where we hope it will be less jarring. ***Which remaining statements are missing citations or imprecise? We want to have this section entirely clear, as it is crucial for the reader to understand our motivation.***
>
> > What does it mean for the training dataset to be “compatible” with either of the two rules?
>
> Your interpretation is correct: Every input either satisfies *both* NESTED *and* EQUAL-COUNT (positive label) or satisfies *neither* NESTED *nor* EQUAL-COUNT (negative label). When we describe a dataset as compatible with a particular rule, it means that the negative and positive labels given are correct for both rules. In other words, there are no training examples which satisfy equation 1 but not equation 2. So both rules classify each sequence in the training data identically, as either true or false. We restate this explanation in our revision to help readers who have not previously encountered such “ambiguous rule” scenarios.

---

> > ### Author Response · Authors · 2025-11-21
> >
> > > Where are the “OOD output probabilities” coming from?
> >
> > We obtain “OOD output probabilities” by running each model on the OOD data and recording the probability assigned to the true label in its output, yielding a vector of outputs assigned by a model to the OOD inputs. We treat these vectors as representations of each model’s respective OOD behavior, and they are used to generate the low-rank visualization of model clusters.
> >
> > > A broad conclusion of this work seems to be that even what we sometimes consider causal in XAI (interpretability ablations) is actually not strictly causal under OOD data.
> >
> > We appreciate the effort that you have put into understanding our paper, but this isn't how we would describe it. We don’t argue that ablations are non-causal, but that model components might not *implement* a particular behavior—as found by causal ablation—and yet may still be used to *predict* that behavior. We focus on OOD cases not as a way of testing the robustness of ablation, but as a way of describing the model’s possible algorithms through its edge case behavior.
> >
> > > For a stronger paper, the authors should make more precise claims, and perhaps better separated claims, about 1) what they can interpret from an attention-based model trained in this synthetic setting where it learns hard rules or heuristics, and 2) what the implications are for current interpretability practices, from the finding that ablation analyses intended to demonstrate mechanistic causality are in fact data-dependent.
> >
> > Now that we have explained the paper in response to your questions, could you be more specific about what sorts of claims you refer to in (1)? As for (2), we hope that the community shifts more towards our goal of predicting model behavior on unseen data, rather than simply steering models. This is an objective that *exploits* data-dependent behavior as a test of understanding, rather than having data-dependence be a *problem* for faithfulness claims.

---

### Official Review · Reviewer_bny3 · 2025-11-01

**Soundness:** 2
**Presentation:** 3
**Contribution:** 2
**Rating:** 2
**Confidence:** 3

**Summary:**

The authors aim to explore whether interpretability can be used to predict OOD generalization for unseen data. They introduce EQUAL-COUNT and NESTED rules into a synthetic parentheses dataset to investigate if interpretations of in-distribution data could predict the OOD behavior on the testing set. I think the topic is quite relevant and important for the community, looking at how interpretability can be used to analyze the model behavior. However, the authors only tested the idea on Transformer models trained on a synthetic dataset, which may not be sufficient to validate the idea reliably.

**Strengths:**

1. Using interpretability to analyze the model behavior could be an interesting topic.
2. The authors provide the dataset and code with detailed experimental settings for good reproducibility.

**Weaknesses:**

1. The title can be misleading. The authors mainly look at the OOD generalization, and only test the miniGPT type transformer model on a specific synthetic dataset. To me, it is not appropriate to use a general phrase “model behavior”.
2. Also, with a transformer-based architecture with different hyperparameters that can influence the OOD generalization, I think it shouldn’t have been stated “hundreds of models”.
3. All the results presented in the paper rely heavily on specific synthetic dataset configurations and the transformer model architecture with limited hyperparameters. I would assume that this method may not be easily applicable to real-world problems. It is unclear whether the key conclusions of the paper will still remain valid when applied to real-world scenarios across different model architectures.
4. The authors mention that a head is a hierarchical head if it tracks depth on at least 80% of mixed-depth inputs. How sensitive is the method to this threshold? The paper did not justify this threshold or investigate the sensitivity.
5. The authors take space to discuss the philosophical motivation. However, I did not see a tight link based on their empirical results.

**Questions:**

1. Did the authors try to adapt the proposed method to real-world problems?
2. Can the authors discuss whether the main conclusions will remain valid in more complex cases?
3. Did the authors consider other quantitative evaluations regarding the OOD generalizability?

---

> ### Author Response · Authors · 2025-11-21
> **response**
>
> Thank you for your constructive review!
>
> > The title can be misleading. The authors mainly look at the OOD generalization, and only test the miniGPT type transformer model on a specific synthetic dataset. To me, it is not appropriate to use a general phrase “model behavior”.
>
> It is true that we are looking at a very restricted toy setting. Our goal is to use it as a case study to describe our novel objective in interpretability and to show that it is a distinct objective from other popular objectives of interpretability, specifically causal mechanistic objectives. We reemphasize this primary goal, and have de-centered and simplified the specific methodological details in our setting.
>
> >  with a transformer-based architecture with different hyperparameters that can influence the OOD generalization, I think it shouldn’t have been stated “hundreds of models”.
>
> Can you clarify what the issue is? We do find that with some hyperparameters, the random seed can introduce large variation, which is why we consider the resulting models to be different models. There are several clusters of OOD behaviors and these clusters don’t cleanly align to single hyper parameter settings, but instead a number of hyper parameter settings can lead to different clusters depending on seed. We do have some additional experiments with higher learning rate as well, although we didn’t initially include them because they don’t always converge, but we are now working to add the ones that do to the experiment pool.
>
> > It is unclear whether the key conclusions of the paper will still remain valid when applied to real-world scenarios across different model architectures.
>
> Thank you for highlighting a point of easy confusion. Our paper is not proposing a specific method for interpreting models, but is proposing an objective—predicting model behavior from interpretations. The specific methods that we use are not important, and only discussed as a demonstrative case study. The key conclusions of the paper are that holistic computational interpretability is a worthwhile goal, and that this goal is decoupled from causal mechanistic interpretability. We believe that these *key conclusions* can easily be applied to real world scenarios as a proposed goal, although the methods used to achieve it will be different.
>
> We are currently also working on adding another task, a synthetic language modeling task with clustered hierarchical generalization behavior documented in https://arxiv.org/abs/2412.04619 which makes it an appropriate setting to look for internal signatures of OOD behavior. We will update the paper once these experiments are complete.
>
> > The authors mention that a head is a hierarchical head if it tracks depth on at least 80% of mixed-depth inputs. How sensitive is the method to this threshold? The paper did not justify this threshold or investigate the sensitivity.
>
> There is a spectrum, where almost all models have at least one head with violation-tracking patterns on 50% of the data. We chose 80% because it happens to sit in the middle of an inflection where a linear trend is disrupted, suggesting possible clustering behavior which was supported by our early experiments. We found that lowering the threshold did not substantially change the main results.
>
> > The authors take space to discuss the philosophical motivation. However, I did not see a tight link based on their empirical results.
>
> See our comment above about our primary goal. We will reemphasize the connection to our empirical results: First, we are able to predict OOD model behavior from intuitions about the model internals. Second, we can predict this behavior from the model internals *even when* those specific internal components are not causally implicated in the behavior being predicted.
>
> > Did the authors try to adapt the proposed method to real-world problems?
>
> Please see our top level comment. The objective of our paper is not to propose a method for prediction, but to propose prediction as an objective, which is novel to evaluations for interpretability. We are running extra experiments to demonstrate that the method---which is not important to our thesis---happens to also work on language modeling tasks in simplified settings where hierarchical generalization is not consistently achieved. However, our primary contribution is not the specific prediction method.
>
> > Can the authors discuss whether the main conclusions will remain valid in more complex cases?
>
> The main empirical finding is that a mechanism's causal validity can be decoupled from its predictive power. This is evidence that these conditions *can* exist and should be taken into consideration in interpretability research, but we do not intend to claim they are guaranteed to exist in any particular situation.
>
> > Did the authors consider other quantitative evaluations regarding the OOD generalizability?
>
> Our specific setting has a very straightforward evaluation of OOD generalizability, which is why we picked it.

---

### Official Review · Reviewer_o6W7 · 2025-11-01

**Soundness:** 3
**Presentation:** 4
**Contribution:** 3
**Rating:** 6
**Confidence:** 2

**Summary:**

The paper investigates whether attention patterns in Transformers can be used to predict how the model will act on Out-of-Distribution (OOD) data. While the experiments show that attention patterns do **correlate** with OOD behavior, they are not necessarily the **cause** of the behavior (attention ablation does not always inhibit the behavior). Finally, the paper cautions that ablation (interventions) on Neural Network architectures should be done on both In-Distribution and OOD data since the results of the ablation can differ considerably.

**Strengths:**

The paper has a great presentation. The Figures are of high quality, the text is easy to read, and the experiments are well explained.

The paper tackles an important challenge: whether we can use attention heads to predict how a Transformer will act "in the wild". Notably, the fact that attention ablation has no effect In-Distribution, but can have unpredictable behaviors Out-of-Distribution is an important result. It warns explainability researchers that only evaluating explainability methods on In-Distribution data does not give the full picture of the model. This observation might influence research in other areas e.g. explainability in computer vision.

While the paper focuses on a simplified setting (simple data and models), it pushes this experimental setup to the limit. The experiments investigate various model depths, weight decays, number of attention heads (Appx C.2). Transformers are also compared with LSTMs in Appx C.1, showing that LSTMs are unable to learn the "Equal-Count" rule.

**Weaknesses:**

## Incomplete Related Work

While Section 2 is a great read, I think it is too high-level for ICLR. I would rather motivate the current work by having a Related Work section that discusses in more depth the papers from the introduction. It would be interesting to describe what is activation steering, activation patching, and Sparse Autoencoder (SAE), and their limitations when it comes to OOD data. For instance, the work of (Kisanne et al. 2024, Smith et al. 2025) (line 48 of the manuscript) focus on limitations of SAEs. Is there existing work that shows limitations of other techniques (activation steering and patching) when it comes to OOD data?

## Citations

Some citations are not correct. For example, the paper "Attention is all you Need" is cited with year 2023 while the paper was published in 2017. Other papers are cited using their Arxiv version while they were published e.g. "Extracting Latent Steering Vectors from Pretrained Language Models" was published at ACL 2022. The citations should be corrected in the final manuscript.

## Confusing Terminology

The paper employs a lot of new terminology which can be hard to follow. Appendix A helped me a lot but it is not referenced in the manuscript. I stumbled upon it by chance. Adding a reference to Appx A would help greatly.

**Questions:**

How sensitive are some of the conclusions to the length of the sequences? Are there some heads that track depth on short sequences but not on large ones?

Are there other ways to perform ablation of attention head? Perhaps forcing a token to only attend to its immediate neighbor is an interesting alternative that inhibits the network from using long-term dependencies.

---

> ### Author Response · Authors · 2025-11-21
> **response**
>
> Thank you for your careful read of our paper! We appreciate the compliments to our presentation, goals, and rigor.
>
> > Incomplete related work
>
> As discussed in our top level response, we prioritize our epistemology and contrast it with currently popular evaluations, which is why we focus on that distinction. We agree that it’s an unusual discussion section for an ICLR paper, but our objective is an empirical “position” with experimental support, so we think it’s important to have our position clearly expressed. However, our revised version does include additional details about the case studies we draw on in the natural sciences and we have moved it to the end of the paper to make it less jarring.
>
> We appreciate the suggestion to include a background section about the robustness of interpretations, although this topic is only crucial to the findings in section 5.3.3. These OOD failures of the *interpretations* certainly relate to our goal, though robust causal interpretation is neither necessary nor sufficient to predict OOD *target* model behavior.  We have added a brief literature review on interpretation robustness to 5.3.3.  Unfortunately, there is limited existing literature on the topic, which is an indicator of the novelty of our work, but we welcome suggestions of any missing citations.
>
> > Citations
>
> We have fixed the reference to Vaswani 2017, which we missed because the bibliography was built directly from arxiv metadata. We have also tracked down the official published metadata for the rest of the bibliography.
>
> > Confusing Terminology
>
> Thank you for highlighting this issue which appears to pose trouble for several reviewers. We have added a reference to the appendix and we have adjusted several terms which seem to be particularly confusing. We also have modified our terminology to avoid forcing the reader to think in terms of tree depth. Rather than negative and nonnegative depth tracking, we now refer to tracking violations and tracking non-violations. Rather than sign-matching head, we refer to a violation-matching head, and a negative depth tracking head is now simply a violation tracking head. We clarify these attention behaviors by emphasizing that heads are checking for violations of the second constraint in Equation (2).
>
> > How sensitive are some of the conclusions to the length of the sequences? Are there some heads that track depth on short sequences but not on large ones?
>
> We define depth tracking as requiring an overwhelming majority of ID cases. Depending on the depth we are talking about, it’s possible we would find more hierarchical heads that only have this behavior for smaller subsets of the data. We have run an additional experiment to see if the hierarchical heads are still predictive for long vs short.
>
> > Are there other ways to perform ablation of attention head? Perhaps forcing a token to only attend to its immediate neighbor is an interesting alternative that inhibits the network from using long-term dependencies.
>
> This is an interesting suggestion. We use uniform attention ablation because in theory, the hierarchical rule can be easily implemented with uniform attention (according to Liu et al.). By contrast, we would not expect to be able to implement in a single layer while zeroing out all tokens except the most recent one. We do try ablating one head at a time as well as across all heads in Appendix G, but we find similar results regardless of which approach we use.

---

> > ### Comment · Reviewer_o6W7 · 2025-11-25
> > **Reviewer Response**
> >
> > Thank you very much for addressing my concerns and responding to my questions. I no longer have any technical issues with the manuscript. Accordingly, I have raised my score to an 8.
> >
> > Issues with some citations remain: arxiv versions are cited instead of published versions (probably because the arvix metadata is not up-to-date). For example
> > - Data distributional properties drive emergent in context learning in transformers was published in NeuRIPS 2022
> > - Causal abstraction: A theoretical foundation for mechanistic interpretability, JMLR 2024
> > - Is this the subspace you are looking for? An interpretability illusion for subspace activation patching, ICLR 2024.
> >
> > I would encourage the author to fixe such reference in the published manuscript.

---

> > > ### Author Response · Authors · 2025-11-25
> > > **Long vs short**
> > >
> > > Our experiments on differences between long and short sequences is complete. We find that the sign of the final token is the only the factor which clearly changes hierarchical attention patterns, and that sequence length doesn’t make a significant difference. Thanks for the suggestion!

---

> > > > ### Comment · Reviewer_o6W7 · 2025-11-26
> > > > **Thank you**
> > > >
> > > > I want to thank the authors for conducting these additional experiments. Adding them to the final manuscript will improve the robustness of the conclusions drawn from the experiments.

---

### Author Response · Authors · 2025-11-21

We thank all reviewers for their thorough efforts. There are two common threads in the weaknesses, which are related. First, some reviewers were surprised by our unconventional decision to start the paper with a philosophical discussion. Second, several reviewers questioned whether our methods or findings will generalize to realistic settings or other tasks.

The core problem is that our goal is outside the norm in interpretability, requiring explicit discussion about how our goal differs and why we consider it worthwhile. In the interpretability field, where questions of what counts as faithful explanation are paramount, we believe that researchers should not shrink away from the philosophy of science.

Our primary objectives are:
- To convey a novel goal (predicting future, unseen model behavior from our interpretations of activations).
- To show that it is possible to achieve this goal in a restricted setting.
- To demonstrate that this goal is distinct from a more popular causal mechanistic objective—and that mechanistic faithfulness is neither necessary nor sufficient to achieve our goal.

Importantly, none of these points rely on any specific method or result generalizing from our toy setting to a real task. Our objective is to illustrate the goal and show that it is not necessarily guaranteed by causal faithfulness. There are several other findings that we discuss, because our setting provides several other intriguing results, but these are the primary objectives. We hope that the reviewers understand that generalization is not a concern for them, despite our discussion of other, more context-specific findings. We would appreciate some feedback on what results we should move to the appendix to avoid confusing distractions. We are also currently running similar experiments in a synthetic language modeling setting which is previously documented to exhibit clustering OOD generalization behavior across random seeds. We hope to show that we can find intuitive links between ID representations and OOD behavior in that setting as well.

As several reviewers noted, it’s unusual to have a philosophical discussion section in an ICLR paper, but our objective is an empirical “position” with experimental support, so we think it’s important to have our position clearly expressed. However, our revised version does include additional details about the case studies we draw on in the natural sciences. Recognizing that several reviewers found it difficult to adapt to our paper structure, we have also moved the philosophical motivation to the end of the paper where a reflective discussion section would normally sit, rather than early in the paper as a motivation section.

Although o6W7 complimented our presentation and explanation, several other reviewers clearly still struggled to follow the central thesis and so focused on our setting-specific method or on minor results. We have revised the written presentation for clarity. Specifically, we have moved some experimental details into the appendix, restated the nature of the toy task in the introduction, clarified the concept of an ambiguous rule experiment, moved some extended philosophical discussion later in the paper, highlighted our terminology reference in the appendix, and changed some terminology to be more comfortable for readers who have not encountered hierarchical generalization research before (moving from “negative vs nonnegative depth” phrasing to clarify that negative depth positions are violations of the condition in equation 2).

---

### Note · Authors · 2026-01-09

I have read and agree with the venue's withdrawal policy on behalf of myself and my co-authors.